# Hedgehog Signaling in Cancer: A Prospective Therapeutic Target for Eradicating Cancer Stem Cells

**DOI:** 10.3390/cells7110208

**Published:** 2018-11-10

**Authors:** Ita Novita Sari, Lan Thi Hanh Phi, Nayoung Jun, Yoseph Toni Wijaya, Sanghyun Lee, Hyog Young Kwon

**Affiliations:** Soonchunhyang Institute of Medi-bio Science, Soonchunhyang University, Cheonan 31151, Korea; itanov07@gmail.com (I.N.S.); hanhlan.11@gmail.com (L.T.H.P.); jny0407@naver.com (N.J.); yoseph.toni@gmail.com (Y.T.W.); october-92@daum.net (S.L.)

**Keywords:** Hedgehog signaling pathway, cancer therapy, cancer stem cells

## Abstract

The Hedgehog (Hh) pathway is a signaling cascade that plays a crucial role in many fundamental processes, including embryonic development and tissue homeostasis. Moreover, emerging evidence has suggested that aberrant activation of Hh is associated with neoplastic transformations, malignant tumors, and drug resistance of a multitude of cancers. At the molecular level, it has been shown that Hh signaling drives the progression of cancers by regulating cancer cell proliferation, malignancy, metastasis, and the expansion of cancer stem cells (CSCs). Thus, a comprehensive understanding of Hh signaling during tumorigenesis and development of chemoresistance is necessary in order to identify potential therapeutic strategies to target various human cancers and their relapse. In this review, we discuss the molecular basis of the Hh signaling pathway and its abnormal activation in several types of human cancers. We also highlight the clinical development of Hh signaling inhibitors for cancer therapy as well as CSC-targeted therapy.

## 1. Introduction

The Hedgehog (Hh) signaling pathway was first discovered in 1980 by Nusslein-Volhard and Wieschaus in a large-scale genetic screening to find mutations that affect larval body segment development in the fruit fly, *Drosophila melanogaster* (*D. melanogaster*) [1]. The Hh signal transduction pathway is an evolutionarily conserved pathway [2] that plays an important role in the regulation of a variety of developmental and physiological processes involving normal stem cell differentiation and proliferation, as well as proper segregation in invertebrates and vertebrates [3,4,5]. However, misregulation of the Hh pathway during embryonic development has been found to cause a variety of birth defects [6,7,8,9,10,11] as well as several neoplastic transformations and malignant tumors [10,12,13,14,15,16,17,18]. Moreover, as the Hh pathway plays an essential role in the maintenance of somatic stem cells and pluripotent cells, it is involved in the regenerative proliferation of epithelial stem cells in the lung [19], tooth [20], liver [21], prostate [22], and bladder [23], and this signaling pathway may be correlated with the maintenance of cancer stem cells (CSCs). Thus, targeting CSCs through inhibition of the Hh signaling pathway is a potential treatment that could effectively improve the clinical outcomes of cancer patients. This review describes the signaling pathway of Hh and its aberrant activation in several cancers and highlights the current clinical trials involving Hh inhibitors for cancer therapy.

## 2. Elements of the Hh Signaling Pathway

Even though there are several important differences between *Drosophila* and other higher organisms, the mechanisms of the Hh signaling pathway are highly conserved [24]. In humans, the Hh pathway has several main components: (1) three Hh homologs, (2) Patched1 (PTCH1 in humans, Ptch1 in mice, and Ptc in *Drosophila*), (3) a G-protein-coupled receptor (GPCR)-like receptor Smoothened (SMO in humans and Smo in mice/*Drosophila*), and (4) three transcription factors (GLI1, GLI2, and GLI3) named from the correlation of GLI1 and glioma [25].

### 2.1. Hh Proteins

Hh proteins are one of the secreted signaling protein families that can be diffused and establish gradients in target tissues after secretion, and thus are responsible for cell interactions during embryonic development [26]. Hh proteins can act as either morphogens or mitogens which control developmental processes during multiple developmental stages in many different tissues in a dose-dependent manner [26]. Hh signaling plays an important role in postnatal bone homeostasis [27,28] as well as in regulating the development and function of theca cells [29] and Leydig cells [30], indicating that Hh ligands not only participate in embryonic patterning but are also involved in the maintenance of tissue homeostasis over the course of a lifetime. Hh proteins undergo maturation by post-translational processing and multiple covalent modifications before the active ligand is released and activates the Hh pathway [27,31]. The N-terminal signal sequence is removed after being translated, and the Hh protein is then autocatalytically cleaved internally [32]. During this process, the C-terminal domain acts as an intramolecular cholesterol transferase to promote the covalent attachment of a molecule of cholesterol to the N-terminal signaling domain [33], while a palmitic acid moiety is added to the N-terminal cysteine residue by the acyltransferase Skinny Hedgehog (Ski) to form a mature Hh protein dually modified by cholesteryl and palmitoyl adducts [34]. Surprisingly, this unusual dual lipid modification plays a vital role in how Hh ligands can move far away from the organizing centers where Hh is specifically expressed and acts as a morphogen [35,36]. Therefore, this dual lipid modification could facilitate the control of the long-range activity of Hh ligands in cancers even where the local Hh proteins are limited [37].

In mammals, there are three Hh gene family members: Sonic hedgehog (SHH), Indian hedgehog (IHH), and Desert hedgehog (DHH). They are all considered to be similar in their physiological effects, but their roles in development are different in many organ systems due to the diversely regulated patterns of expression [1]. Of all the Hh ligands, SHH is the most studied ligand and acts as a cell–cell signaling factor, modulating cell fates through an autocrine or paracrine mechanism. During embryogenesis, SHH has been shown to be expressed in embryonic structures and involved in pattern formation in the embryo, including the development of the limb at the zone of polarizing activity (ZPA), notochord, or the floor plate of the neural tube [38,39] In addition, SHH is also expressed in the development of the lung [40,41], teeth [42,43], brain [44], hair follicle [45], pancreas [46], and intestine [47]. IHH is involved in the differentiation and cell fate decision of several specific tissues, including the differentiation of colonic epithelial cells [48,49] and the decision of neuroectodermal cell fate by activating the development of the earliest hemato-vascular system [50]. It is implicated in bone formation due to its role in regulating the rate of chondrocyte differentiation [51,52] and promoting human cartilage endplate (CEP) degeneration [53]. Interestingly, SHH [54,55] and IHH [56] were shown to be involved in the commitment and differentiation of the T cell lineage, as well as the proliferation and survival of developing T cells. It was also reported that SHH regulates the B-lineage commitment of hematopoietic progenitor cells and the development of B cells [57]. Whereas SHH and IHH are closely related to each other, DHH is the closest homolog to *D. melanogaster* of all the discovered Hh ligands [58]. DHH expression is mainly restricted to gonads, such as Sertoli cells [58] and Leydig cells [30] in the testis and granulosa cells of growing follicles in the ovaries [29], where it plays an important role in gametogenesis and steroidogenesis. Besides this, DHH could negatively regulate erythrocyte differentiation at multiple stages in both the spleen and bone marrow [59].

### 2.2. PTCH

The Hh/SHH receptor is PTCH [60,61], a 12-pass transmembrane protein that has two large extracellular loops and two large intracellular loops [62,63]. Two mammalian PTCH homologs have been identified: Patched1 (PTCH1) and Patched2 (PTCH2). It was shown that they bind the three Hh ligands with equal affinity and inhibit the activity of the SMO protein [18]. While PTCH1 is primarily expressed in mesenchymal cells throughout the embryo and plays a role as the primary mediator for most SHH activities, PTCH2 is specifically expressed in skin cells and spermatocytes; it is therefore likely to participate in the function of DHH in germ cells as DHH is mainly expressed in the testis [64]. Mutations of the *Patched* gene have been demonstrated in several diseases such as basal cell nevus syndrome (BCNS), nevoid basal cell carcinoma syndrome, sporadic basal cell carcinomas, and medulloblastomas [65,66,67].

### 2.3. SMO

SMO is a seven-pass integral membrane protein that is a member of the Frizzled (FzD) class of G-protein-coupled receptors (GPCRs) and functions as a positive regulator of the Hh signaling pathway because of its physical characteristics and position in Hh signaling by acting downstream of or in parallel to Patched [68]. SMO has an extracellular cysteine-rich domain (CRD), which binds to small-molecule modulators and is therefore indispensable for SMO function in the Hh signaling pathway [69]. It has been indicated that SMO does not directly bind SHH [70]; Hh binds specifically to PTCH without any help from SMO and consequently promotes the conformational change resulting in the releasing of SMO [71]. Moreover, SMO can form a physical complex with PTCH1, which indirectly inhibits SMO activity [61]; the mechanism is still not clear, but possibly involves changes in the distribution or concentration of a small, unknown molecule [72]. In addition, SMO is induced by Hh through the phosphorylation by protein kinase A (PKA) and casein kinase I (CKI), which regulate its cell-surface accumulation and signaling activity [73].

### 2.4. GLI

The human *Gli* gene is located at chromosome 12 (q13 to q14.3) and was identified by Vogelstein in 1987 because of its gene amplification of more than 50-fold in glioblastoma multiforme (GBM) and its derived cell line [74]. In mammals, three members of the Gli gene family have been identified—GLI (or GLI1), GLI2, and GLI3, which have five successive repeats of highly conserved zinc finger DNA-binding domains, characterized as members in the Kruppel family of zinc-finger-containing transcription factors. Moreover, they require the carboxyl-terminal amino acids 1020–1091, which include an 18-amino-acid herpes simplex viral protein 16-like activation domain, to act as transcription factors in the vertebrate SHH–Patched signaling pathway [75]. These findings support the hypothesis that GLI proteins are the terminal evolutionarily conserved transcription factors of the Hh signaling pathway and directly bind to the promoters of their target genes [76]. After being translated, GLI proteins mainly undergo nuclear localization and bind their DNA binding site with high affinity to protect a 23- to 24-bp region, including the 9-base-pair consensus sequence 5′-GACCACCCA-3′ [77].

Among the three GLI family members, the *Drosophila*
*Ci* is more closely functionally related to mammalian GLI2 and GLI3, as they can activate transcription and undergo proteolysis to generate both activating and repressing domains, whereas GLI1 cannot be modified post-translationally and thus only plays a role as a transcriptional activator of the Hh signaling pathway [78]. GLI1 has been demonstrated to be a mediator of the SHH signal in vertebrates [79] by binding to the promoter region of several target genes involved in different cellular processes, such as G1 cell cycle progression, tumor formation, and progression [80]. On the other hand, GLI2 primarily acts as a transcriptional activator (GLIA) with activity in patterning the ventral regions of the spinal cord, whereas GLI3 mainly functions as a transcriptional repressor (GLIR) of the Hh signaling pathway, and plays a role in patterning the intermediate spinal cord [81]. Mechanistically, the three mammalian GLI proteins share a similar DNA-binding domain including five zinc finger repeats and an activation domain in the C-terminal, but only GLI2 and GLI3 have the repression domain at the N-terminus even though GLI2 acts only as a weak transcriptional activator [82,83,84].

## 3. Cascades of Canonical Hh Signaling

In the absence of an Hh ligand, PTCH1, a 12-pass transmembrane receptor, localizes at the base of the primary cilium (PC), a subcellular membrane extension that is important for the distribution and function of most of the components in this pathway. PTCH1 then destabilizes SMO by repressing its ciliary accumulation and consequently inhibits SMO activity, which is an essential step for activating this pathway in mammals (but not in *Drosophila*) [85]. The full-length GLI is therefore sequentially phosphorylated at multiple sites in the C-terminal region by protein kinase A (PKA), glycogen synthase kinase-3 (GSK3), and casein kinase 1 (CK1), [86]. The phosphorylated GLI protein directly binds β-transducin-repeat-containing protein (β-TrCP) in the Skp1-Cul1-F-box protein (SCF) ubiquitin–ligase complex via two binding sites; after that, it is subsequently ubiquitinated and degraded by the proteasome [87]. The truncated N-terminal GLI repressor (GLIR) then translocates to the nucleus and binds to Hh target gene promoters and switches off their expression in a sustained manner [88].

In the canonical Hh signaling pathway, Hh regulates the GLI family of transcription factors. Upon Hh binding to PTCH1, PTCH1 leaves the cilia by internalizing together with its ligand SHH, and both proteins are degraded in lysosomes [89]. This binding therefore relieves the inhibition of SMO. SMO is then accumulated and translocated into the PC [90]. The Hh signaling is subsequently activated and transmitted via a protein complex, which includes kinesin protein (Kif7) and Suppressor of fused (Sufu). In the absence of Hh, Sufu restrains GLI protein in the cytoplasm, whereas binding of Hh to its receptor activates the release of GLI from Sufu and allows it to enter the nucleus [91]. In addition, Kif7 has also been involved in the regulation of the Hh signaling pathway as a ciliary motor in which the activation of the Hh pathway promotes its ciliary trafficking from the base to the tip of the PC [92]. In the absence of Hh, Kif7 accumulates at the base of the PC, and Kif7 physically interacts and prevents GLI protein accumulation within the cilia by controlling their proteolysis and stability [93]. Upon Hh binding, Kif7 localizes at the distal tip of the PC and prevents the cleavage of GLI3 into a transcription repressor [94]. In the end, GLIFL is then converted to its active form GLIA and migrates to the nucleus to activate the expression of a number of target genes including *PTCH1* [95]. A schematic of the activation of the canonical Hh signaling pathway is shown in Figure 1.

## 4. Noncanonical Hedgehog Pathway

However, emerging evidence over the past few years has indicated that Hh proteins can also signal through GLI-independent mechanisms, the so-called “noncanonical” Hh signaling [5]. In this pathway, all the responses to Hh ligands are not through transcriptional changes mediated by the transcription factors of the Gli family [96]. Interestingly, Hedgehog signaling in CSCs is noncanonical PTCH1-dependent, which is a positive regulator of WNT signaling, whereas canonical SMO-dependent Hedgehog signaling is involved in the downregulation of WNT signaling in normal and differentiated cancer cells through nuclear localization of GLI1 [97]. Noncanonical Hh signaling has been classified into two types: (1) SMO-independent and exclusively through PTCH, and (2) SMO-downstream, mediated by activation of small GTPases, but independent of GLI transcription factors. The SMO-independent GLI activation pathway has additionally been shown to be noncanonical Hh signaling [98,99].

The SMO-independent noncanonical signaling pathway, which operates via PTCH and three Hh proteins (SHH, IHH, and DHH), can inhibit the activation of caspase-3 and promote cell survival through inactivating the PTCH1 pro-apoptotic activity in an SMO-independent manner [100]. Surprisingly, it is indicated that the autocrine Shh–Ptch–Smo signaling is not required in pancreatic ductal cells for the progression of pancreatic ductal adenocarcinoma (PDAC), as the deletion of SMO in the pancreas does not affect the multistage development of PDAC tumors [98]; instead, GLI1 is regulated by TGF-β and KRAS, and Gli–NF-κB oncogenic activation is required for the Kras-dependent transformation of cultured PDAC cancer cells [101]. Besides this, Smo- and Gli-independent Hedgehog also participate in the regulation of the cell cycle through Cyclin B1 and G-protein receptor kinase-2 (GRK2) [102].

In the noncanonical signaling pathway, which works through functions or activities of SMO, it has been indicated that via the rapid stimulation of Rac1 and RhoA small GTPases by SMO, heterotrimeric Gi Proteins are required for SHH-induced fibroblast migration in human endothelial cells [103]. Additionally, cholangiocarcinoma cells exhibit a GLI- and cilia-independent noncanonical Hh signaling pathway, which contributes to mammalian tumor cell engraftment and chemotaxis, including cytoskeleton remodeling and cell migration [104]. Moreover, arachidonic acid (ARA) metabolism through the lipoxygenase pathway plays an important role in Gli-independent Hh signaling in fibroblast migration [105] and neuronal development, particularly in the synthesis of leukotrienes [106]. In addition, the noncanonical Hh signaling pathway is also involved in controlling axon guidance in a Smo-dependent manner by inducing phosphorylation and activation of Src family kinases (SFKs) to alter axon trajectories [107].

## 5. Types of Aberrant Activation of Hedgehog Signaling in Diseases and Cancers

Accumulating evidence suggests that aberrant activation of the Hh signaling pathway by deregulation of any component within may be involved in the development and progression of cancers and diseases. In various cancer types, three types of proposed mechanisms of uncontrolled activation of Hh signaling are explained below, as summarized in Table 1.

### 5.1. Type I—Ligand Independent

The first evidence of a link between aberrant Hh signaling and cancer is the rare condition Gorlin syndrome (also known as Naevoid Basal Cell Carcinoma Syndrome or Basal Cell Naevus Syndrome), which is caused by the activating mutation in the *Patched* gene [108]. Gorlin syndrome is characterized by developmental abnormalities and a distinct postnatal occurrence of cancers that are also known to be caused by the aberrant activation of Hh signaling [66,109]. Cancers in patients with Gorlin syndrome include basal cell carcinomas (BCCs) [66,110], the most common cancer in the Western world; medulloblastoma (MB) [111,112], the most frequent malignant pediatric brain tumor; and rhabdomyosarcoma (RMS), the most common type of soft tissue cancer in children [113].

The tumorigenesis is ligand independent among all these cases, in which the pathway is constitutively activated in the absence of the ligand through mutations in components of the Hh pathway, including activating mutations in the SMO and inactivating mutations in the PTCH1 or Sufu. In Gorlin syndrome, a PTCH1 mutation on chromosome 9 was found [114]. Interestingly, either mutational inactivation of PTCH1 (approximately 30% of cases) or activating mutations of SMO are identified in sporadic BCCs [65,114,115]. In addition, children with MB carry inactivation mutations in PTCH1 or Sufu [112,116,117]. Mutations in Sufu are also found in sporadic BCC, although rarely [118]. Moreover, GLI2 amplifications are found in children (older than 3 years old) with MB as well [119]. Similar to MB, inactivation mutations in either PTCH1 or Sufu are involved in the development of RMS [113]. Apart from these diseases, loss of heterozygosity or somatic mutations in the *PTCH1* gene are observed in invasive transitional cell carcinoma of the bladder [120], esophageal squamous cell carcinoma [121], and trichoepitheliomas [122].

These observations have been confirmed by experiments in a variety of preclinical and clinical models. As heritable mutations in BCNS patients and a somatic mutation in a sporadic BCC were found in the *Ptch1* gene [66], Ptch1 heterozygous knockout mice developed BCC after being induced with ultraviolet and ionizing radiation or through an inducible Keratin6 promoter [123,124]. In addition, BCC-like skin abnormalities were identified in a Smo constitutively overexpressing skin-specific transgenic mouse model, as activating somatic missense mutations in the Smo gene were found in sporadic BCCs [115]. Moreover, mice carrying heterozygous loss-of-function mutations in the *Sufu* gene also develop a skin phenotype similar to that in Gorlin syndrome such as basaloid changes and jaw keratocysts [125].

### 5.2. Type II—Ligand-Dependent Autocrine/Juxtacrine Signaling

Besides activation via the mutations in components of the Hh pathway without ligands, Hh signaling could also be ligand-dependent autocrine/juxtacrine-activated, in which the Hh ligand is profusely released and taken up, activating the same or surrounding tumor cells. In addition to the overexpression of the Hh ligand, other tumors display a high level of PTCH1 and GLI expression.

As the Hh ligand is known to be widely expressed in the foregut endoderm and involved in the regulation of branching morphogenesis in the mammalian lung [126], Hh signaling is proven to participate in the malignant phenotypes of airway epithelial progenitors and small-cell lung tumors, and the first Hh-overexpressing tumor was identified [27]. The activation of the Hh signaling pathway in a ligand-dependent manner was later observed in a variety of different cancers, including digestive tract [127], colorectal [128], prostate [129], liver [130], breast [131], ovarian [132], and brain [133] cancer and melanoma [134]. To prove the autocrine mechanism of the Hh signaling pathway, an Hh-neutralizing antibody or exogenously added Hh ligand was used [127]. Moreover, cyclopamine, a Hedgehog pathway inhibitor, also enhanced the apoptosis and suppressed the proliferation of tumor cells as well as the growth of tumor xenografts after treatment, which strengthened the hypothesis that the Hh pathway could be activated via Hh ligands released by the same or surrounding tumor cells [128,135,136].

### 5.3. Type III—Ligand-Dependent Paracrine Signaling

In addition to autocrine/juxtacrine activation of the Hh signaling pathway, the Hh pathway can be activated through a paracrine manner in stromal cells as well. In this type, Hh ligands released by cancer cells will bind to the PTCH1 receptor and switch on the Hh signaling in stromal cells. These stromal cells will secrete paracrine growth signals such as vascular endothelial growth factor (VEGF), insulin-like growth factor (IGF), interleukin-6 (IL-6), Wnt, Platelet-derived growth factor (PDGF), and Bone morphogenetic proteins (BMP) to induce the growth of the tumor [12]. The paracrine activation of the Hh pathway has been shown in several cancers such as prostate [137], pancreatic [138], and colorectal cancers and in subsets of esophageal cancers [139]. For instance, in prostate cancer specimens, the expression of SHH is also detected in the tumor epithelium while GLI1 expression is found in the tumor stromal cells, which suggests that Hh signaling is induced in stroma cells by tumor cells to induce the growth of the tumor through secreting paracrine signals [137]. Similarly, the SHH transcript is localized to tumor tissue whereas GLI1 and PTCH1 are detected in both the tumor and the stroma, proving the hypothesis that SHH and its target genes function in a paracrine manner to regulate the development of subsets of esophageal cancers [139].

Moreover, this type of Hh signaling pathway could also be in a reverse paracrine manner, in which the cancer cells take up the Hh ligands released by stromal cells and promote their growth. For example, Hh ligands released by bone-marrow, nodal, and splenic stroma could activate Hh signaling, which plays an important role in the survival of B and plasma cell malignancies [140]. Moreover, the source of the SHH ligand in the activation of Hh signaling pathway for the pathogenesis of multiple myeloma was also found from bone marrow stromal cells [141]. Interestingly, it was also demonstrated that intact SHH-producing microenvironments or neurosphere conditions were required for GLI activation in gliomas [142].

## 6. Hedgehog Signaling in Cancers and Its Inhibitors

As previously mentioned, while a controlled Hh signaling pathway contributes to numerous biological processes throughout embryonic development of normal tissues and organs as well as to tissue repair and tissue homeostasis, uncontrolled activation of the pathway may promote tumorigenesis [143], such as medulloblastoma, rhabdomyosarcoma, basal cell carcinoma, and lung, colon, stomach, pancreas, ovarian, breast, and malignant hematological cancers [10,12,13,14,15,16,17,18]. Moreover, accumulating evidence suggests that Hedgehog signaling plays a role in the properties of CSCs in several cancers. For instance, the expression of Hedgehog signaling components such as PTCH1, GLI1, and GLI2 is up-regulated in normal human mammary stem/progenitor cells of mammospheres and the Hh signaling pathway is induced in human breast CSCs [144]. Also, the Hh pathway is critical for the maintenance and self-renewal of normal and neoplastic stem cells of the hematopoietic system, including CSCs in myeloid leukemia [145], as well as in solid cancers such as melanoma and pancreatic cancer [146,147]. Some cancer cells surviving after chemotherapeutic treatment such as gemcitabine harbor stem-cell-like properties and display characteristics associated with epithelial–mesenchymal transition (EMT) [148]. Surprisingly, Hh signaling also controls the expression of ABC transporter proteins such as multi-drug resistance protein-1, leading to the chemoresistance of CSCs [149]. Besides this, as CSCs are known to undergo EMT, Hh pathway inhibitors such as cyclopamine and GANT-61 [150] or panobinostat [151] can be used as a potential novel treatment strategy to suppress the EMT process and distant metastasis in cancers. Interestingly, recent evidence has indicated that Hh signaling plays an essential role in tumor angiogenesis through VEGF-A induction in stromal perivascular cells [152], particularly in triple-negative breast cancer [153]. Thus, many inhibitors of the Hh pathway have been developed and tried clinically with promising results (Table 2 and Table 3). However, emerging data indicate that cancer cells may also develop resistance to Hh inhibitors [154,155,156,157,158,159]. For example, in medulloblastoma and BCC, treatment with vismodegib (GDC-0449) and sonidegib (LDE-225), two drugs that have shown significant results in clinical trials, caused acquired drug resistance in the residing cells of these cancers [154,155]. Using genomic analysis, Sharpe et al. indicated that acquired-drug-resistant cells employed SMO mutations in BCC, which further drove drug resistance to SMO inhibitors [155]. The SMO-D4738H mutation was found in 42.5% of BCC patients that acquired drug resistance [156]. This mutation blocked the binding of vismodegib and sonigedib to the mutant SMO protein [156]. While the new generation of SMO inhibitors is undergoing clinical trials, a recent study by Li et al. demonstrated that benzimidazole derivates named HH-1, HH-13, and HH-20 are potent novel SMO inhibitors [160,161,162,163]. Moreover, HH-13 and HH-20 showed prompt inhibition of acquired-drug-resistance SMO-D37738H cells as well [163]. In this section, we discuss the importance of Hh signaling in several cancers, including how Hh signaling affects the survival, EMT, metastasis, and acquired drug resistance of cancer cells and CSC expansion. Also, we discuss the development of the Hh inhibitors that are being clinically trialed to eradicate bulk tumors and CSCs, as summarized in Table 2 and Table 3.

### 6.1. Basal Cell Carcinoma (BCC)

Basal cell carcinoma (BCC) is the first group of cancers where the oncogenic potential of deregulated Hh signaling was identified. The pioneer study by Hanh et al. and Jonhson et al. indicated that patients with Gorlin’s syndrome had a strong tendency to develop BCC [66,108]. It is believed that the upregulation of Hh signaling is the ace and driver of all BCC malignancies. Hh signaling in the BCC belongs to the ligand-independent type I Hh signaling [66]. About 90% of BCC patients have an identifiable loss-of-function mutation in at least one allele of PTCH1, and the remaining 10% have activating gain-of-function mutations in SMO [115]. In line with these findings, an in vivo study using various models of BCC showed that dysregulation of the Hh signaling pathway mediated uncontrolled proliferation of basal cells [115,124,186]. Besides its function in the promotion of tumorigenesis, Hh signaling has also been reported to be involved in the BCC tumor microenvironment, especially cancer immunity [187]. A study by Otsuka et al. showed that Hh signaling inhibition led to enhanced adaptive immune responses in BCC [187]. Hh components were known to be highly expressed in the thymus, and its upregulation led to the inhibition of T cell activation [188,189,190]. IL-4 upregulation in the tumor microenvironment was reported to enhance tumorigenesis as well as inhibit antitumor response [191,192]. Furmanski et al. also showed that IL-4 is a transcriptional target of Hh signaling in T cells [193]. In addition, Hh signaling inhibition was also shown to increase interferon-gamma (IFNγ) expression [193,194]. Taken together, this suggests that targeting Hh signaling in BCC would increase the effectiveness of cancer therapy by modulating the antitumor activity of the immune cells.

Cyclopamine, an Hh signaling inhibitor, is the first drug that has been used to treat BCC [195]. Since then, several other synthetic cyclopamine derivatives have continuously been developed as Hh pathway inhibitors, with better pharmacological and inhibitory properties than those of cyclopamine. Cur-61414, one of the synthetic SMO inhibitors, potentially prevented the formation of BCC-like cells and BCC-like lesions without affecting surrounding normal cells [17,196]. Because of its safety, Cur-61414 was considered for preclinical models and was the first class of Hh antagonist to enter phase I clinical trials for BCC patient treatment [196]. However, it failed to either meet the patient’s response or reduce GLI1 expression [196]. Vismodegib (GDC-0449) was then developed as a second SMO inhibitor with a better ability to inhibit BCC progression and with more favorable pharmaceutical properties than cyclopamine. Vismodegib showed great antitumor activity in preclinical models [158,197,198]. A phase I study of vismodegib that demonstrated antitumor activity in patients with BCC and medulloblastoma was conducted in 2009 [158,197]. Based on these promising results, vismodegib has recently entered phase II trials in advanced BCC. Another SMO inhibitor that also entered clinical trials is sonidegib (LDE225), which is used for treating recurring BCC after surgery and radiation therapy [197]. In addition, saridegib (IPI-926) is another SMO inhibitor that has tolerability in advanced solid tumors, including BCC [199]. Despite the successes of these drugs, it is likely that many tumors acquire clinical resistance during therapy [200].

CD200^+^CD45^−^ cells have been identified as a CSC population in BCC. This population was resistant to chemotherapy and expressed a high level of multidrug resistance protein 1 (MDR1), whose expression gradually increased upon drug exposure. CD200^+^ BCC also expressed a high level of GLI1 and relied on GLI1 for their survival, but not GLI2 [164,201]. In an in vivo study, Colmont et al. found that CD200^+^CD45^−^ cells were able to initiate BCC tumor growth with identical histology to the parental cells and showed the upregulation of SHH signaling genes. Inhibiting Hh signaling by Smo antagonists was successful in eradicating BCC CSCs. Thus, this suggested a hypothesis that dual therapy using available anti CD200^+^ neutralizing antibody in combination with Smo antagonists would be a novel strategy for a complete inhibition of BCC progression [164].

### 6.2. Colon Cancer

The Hedgehog (Hh) signaling pathway is essential for cell growth as well as the development of gastrointestinal tracts [202]. Accumulating evidence suggests that Hh signaling plays important roles in the development of colon cancer [203,204]. Hh signaling in the CRC belongs to the ligand-dependent type II Hh signaling [128]. Even though each colorectal cancer (CRC) has a different pattern of Hh signaling expression, most of them highly express the Hh mRNA or protein [49]. Moreover, the expression levels of PTCH and SMO were gradually increased as the colon cancer progressed from a benign to more malignant tumor [203]. In addition, other components of Hh signaling, such as SHH and GLI1, were also reported as being upregulated in colon cancer [205]. Dysregulation of Hh signaling contributed to gene mutation, metastasis, and angiogenesis in colon cancer [206], and termination of Hh signaling through GLI1 inhibition resulted in the inhibition of proliferation in colon cancer [207,208]. Considering the importance of Hh signaling in the progression of colon cancer, cyclopamine appeared to inhibit the Hh signal and suppressed the survival and proliferation of colon cancer. Cyclopamine treatment resulted in the inhibition of Hh, SMO, and PTCH mRNA, which were highly expressed in the colon cancer cell lines [209]. In addition, cyclopamine inhibited E-cadherin expression in benign or malignant colorectal tumor cells, and reduced their invasive capacity [210]. Vismodegib (GDC-0449), which has been approved for treating BCC, was also utilized for Hh-targeted therapy for colon cancer patients [12,25]. However, drug resistance to vismodegib was acquired over time. In addition, side effects were observed in the patients that underwent vismodegib treatment [25,155,158]. Therefore, a recent study by Chen et al. suggested that Hh003, an SMO antagonist with a new scaffold, induced a greater effect on the elimination of human colon and pancreatic cancer compared with vismodegib. Hh003 exerted stronger antitumor effects compared to vismodegib by activating caspase8-dependent apoptosis [211].

It has been reported that Hh signaling interacts with other signaling pathways, such as Wnt/b-catenin and PI3K pathways in several cancers, including in colon cancer [212]. Dual targeting of Wnt/b-catenin and Hh-GLI1 signaling resulted in a better elimination of colon cancer cells compared with inhibition of GLI1 alone [97,212]. In addition, dual targeting of GLI1 by arsenic trioxide and PI3K inhibitor LY294002 synergistically downregulated both the GLI1 protein level and GLI1 target genes *BCL2* and *CCND1*, which resulted in decreased proliferarion of colon cancer cells [213].

CD133^+^ was identified as a marker of colon CSCs, and Hh-GLI1 signaling was also reported to play important roles in colon CSC survival and expansion [128]. Treatment with cyclopamine showed a reduction in colon cancer progression and metastasis, and inhibition of Hh signaling through GLI1-targeted shRNA consistently displayed dramatically lower colony forming ability in colon CSCs [128]. A study by Batsaikhan et al. showed that the expression of EMT markers and stem cell markers was higher in the colon cancer spheres than in the parental cell line. Treatment of colon CSCs with cyclopamine decreased the expression of SHH and its downstream targets in a dose-dependent manner [214]. Noncanonical PTCH-1-dependent Hh signaling is required for the survival of colon CSCs, especially by its regulation in the activation of the Wnt/b-catenin signaling pathway. It acts as a positive regulator of Wnt/b-catenin signaling to maintain colon CSCs in a quiescent state [97,212]. Treatment of colon CSCs with PTCH1 inhibitor RU-SKI 43 significantly decreased Wnt activity, whereas vismodegib treatment failed to achieve the same outcome [97].

### 6.3. Breast Cancer

Aberrant activation of Hh signaling has been observed in the several types of breast cancers. As we discussed earlier, Hh signaling alteration in cancer arises due to reasons such as overexpression of Hh ligands, which results in the activation of either autocrine or paracrine signaling, loss of function of the receptor, somatic mutation of Hh components such as GLI1 and PTCH1, and the overexpression of the Hh components [215]. In breast cancer, Hh signaling is dependent on the autocrine manner of the Hh ligands of the tumor cells (type II Hh signaling), which in turn acts as a self-activating signal for the Hh amplification [131]. The receptor loss of function normally occurs in PTCH receptors, whereas gain of function was shown mostly in SMO receptors [216,217]. SMO and GLI1 were highly expressed in triple negative breast cancer (TNBC), and their excessive expression was correlated with metastasis, poor prognosis, and recurrence of TNBC [218]. Moreover, in breast cancer, Hh signaling has been shown to induce angiogenesis independently of VEGF activation [219]. Recently, a comprehensive immunohistochemistry (IHC) study by Kurebayashi et al. showed that SHH expression was positively correlated with tumor size, Ki-67 labeling index, and HER2 positivity, but negatively correlated with estrogen receptor (ER+) and progesterone receptor (PR+) breast cancer [220]. In addition, a GLI1 isoform called truncated GLI1 (tGLI1) has also been reported to induce the activation of genes related to proliferation, migration, and angiogenesis of breast cancer. Even though the tGLI1 mechanism of action was different from that of GLI1, it functioned similarly in promoting breast cancer growth [197].

Hh inhibitor sonidegib was used in the treatment of several cancers and has been used clinically to treat breast cancer patients, either independently or in a combination with docetaxel [221]. Cyclopamine treatment inhibited PTCH1 expression and cell cycle progression in breast cancer cells [222]. GLI1 inhibitors such as pirfenidone and imiquimod have been reported to have promising antitumor activity in breast cancer [223,224]. Topical application of imiquimod was well tolerated and decreased the cancer progression [224]. GANT61 is a small GLI1 antagonist that was also reported to have antitumor activity in breast cancer. GANTS61 had high specificity to reduce GLI1 expression and blocked the binding of GLI to the target DNA sequences [225]. Breast cancer treatment with GDC-0449 and LDE225 SMO inhibitors has also been studied in clinical trials [197].

CD44^+^/CD24^−^/Lin^−^ is a subpopulation of breast CSCs that has the ability to produce large numbers of tumors having self-renewal function. CSCs expressing CD44^+^/CD24^−^/Lin^−^ have increased PTCH1, GLI1, and GLI2 mRNA expression [144,226]. P63, a sister homolog of p53, which has been reported to interact with Hh signaling, is also highly expressed in breast CSCs compared to normal cells [216]. In addition, GLI1 is highly expressed in breast CSCs, and a study in an orthotopic xenografts model showed that low expression of GLI1 reduces tumor growth, self-renewal, motility, and viability of breast CSCs [227,228]. Hh signaling inactivation in breast CSCs downregulates the representative stem cell markers OCT4, NESTIN, and NANOG. Through in vitro study, cyclopamine showed the capacity to inhibit mammosphere formation of mammary gland stem cells [144,165].

### 6.4. Pancreatic Cancer

Pancreatic ductal adenocarcinoma (PDAC) is thought to be one of the leading causes of cancer death with 5-year survival rates of 1 to 5% [229]. It has been reported that uncontrolled activation of Hh signaling is efficient in maintaining tumor growth and progression of the pancreatic tumor [230]. Using Pdx–Shh mice, in which sonic hedgehog (Shh) overexpression was driven by the pancreatic-specific Pdx1 promoter, a study by Thayer et al. showed that mouse pancreas developed abnormal tubular structures similar to the phenotype of human pancreatic intraepithelial neoplasia (PanIN-1 and -2) [135]. Not only genetic mutation but also epigenetic modification is involved in pancreatic cancer tumorigenesis. Hedgehog-interacting protein (HHIP), a negative regulator of Hh signaling which is normally downregulated or absent in pancreatic cancer, was considered to be the target of genetic and epigenetic modifications. The HHIP promoter underwent partial and complete methylation in several pancreatic cancer cell lines, primary pancreatic cancers, and pancreatic cancer xenografts, whereas no methylation was found in normal pancreata. This suggests that methylation of the HHIP promoter contributes to the upregulated Hedgehog signaling in pancreatic neoplasms [231].

Hh signaling in pancreatic cancer belongs to type III ligand-dependent signaling [98]. Among the components of Hh signaling, it has been reported that deletion of SMO, a component of SHH–PTCH1–SMO autocrine signaling, did not affect multistage tumor development of PDAC [98]. Furthermore, a study by Yauch et al. showed a ligand-dependent activation of the Hh pathway in the stromal microenvironment of pancreatic cancer [198]. They showed that inhibition of the Hh pathway using a Hh pathway antagonist downregulated the canonical Hh target genes in the stromal microenvironment and inhibited pancreatic tumor growth in xenograft models [198]. Taken together, this suggests that the paracrine ligand for Hh signaling but not autocrine signaling is required for PDAC tumor progression in pancreatic ductal cells [98].

Various studies have shown the ability of cyclopamine to induce apoptosis in a variety of tumor cell lines and to inhibit pancreatic cancer [135,232]. Cyclopamine not only induced apoptosis and inhibited proliferation in pancreatic cancer cell lines both in vitro and in vivo, but also inhibited systemic metastases in spontaneously metastatic orthotopic xenograft models [233]. Similarly, IPI-269609, a Hh-targeted small molecule, had a similar effect to that of cyclopamine in inhibiting pancreatic cancer progression [135,232]. It has been shown that Hh signaling inhibition by cyclopamine significantly extended median survival in the transgenic mouse model [233]. Hh inhibition by cyclopamine led to the down-regulation of SNAIL, up-regulation of E-CADHERIN, and inhibition of the metastasis capability of pancreatic cancer [232]. Indeed, a combination of gemcitabine and cyclopamine showed a complete suppression and significant reduction in the primary tumor size [232]. However, the follow-up to the clinical trials did not show a significant effect. On the other hand, a phase II clinical trial of a IPI-926–gemcitabine combination was closed early because of the side effects of the treatment [234]. The combination of vismodegib with gemcitabine did not improve the response rate and overall survival of patients with metastatic pancreatic cancer. Also, preclinical and clinical results reported by Catenacci et al. showed that the combination treatment did not have a significantly different outcome from that of vismodegib treatment alone [235].

In pancreatic cancer, CD44^+^CD24^+^ESA^+^ cells were identified as a population of pancreatic CSC [232]. Hh signaling is known to be a major regulator of pancreatic CSC self-renewal [169]. Kim et al. showed that even though GDC-0449 led to a downregulation of GLI1 and PTCH1, it did not show any significant differences in the elimination of pancreatic CSCs. Moreover, GDC-0449 and gemcitabine combination therapy did not significantly abrogate the metastasis of pancreatic cancer more efficiently than gemcitabine alone. Therefore, development of a pancreatic CSC-targeted Hh inhibitor still needs to be explored [236]. Li et al. found that treatment with sulforaphane significantly reduced tumor growth and self-renewal of pancreatic CSCs both in vitro and in orthotopically implanted human-derived primary pancreatic CSCs [169]. Inhibition of Hh–GLI1 signaling by GANT-61 (GLI1 antagonist) showed an inhibition of pancreatic CSC growth in both in vitro and in vivo models [166]. Baicalein, an active compound in the QYHJ formula, also showed a great ability to impair pancreatic CSC self-renewal through the inhibition of the SHH signaling pathway [171]. Sanguinarine is another pancreatic-CSC-targeted drug whose usefulness has been challenged. Sanguinarine inhibits the SHH–GLI1 pathway, resulting in the downregulation of GLI1 target genes in pancreatic CSCs. It inhibits the proliferation and self-renewal of pancreatic CSCs by inducing oxidative damage, leading to cell apoptosis [170]. Gemcitabine-resistant pancreatic cancer cells displayed high expression of Hh signaling components as well as CSC markers [172]. Treatment of these resistant cells with cyclopamine reverted the acquired chemoresistance and decreased the expression of CSC markers [168,172]. In addition, treatment with cyclopamine derivate (CyT) along with focal irradiation showed an additive effect on tumor sphere formation in vitro, as well as inhibition of lymph node metastasis in an in vivo model [167]. Taken together, targeting Hh signaling in pancreatic CSCs showed a better outcome in the ablation of pancreatic cancer cells.

### 6.5. Medulloblastoma

Medulloblastoma, an aggressive childhood tumor of cerebellar origin, is a malignancy exhibiting a well-recognized reliance on aberrant Hh signaling. The Hh pathway of medulloblastoma belongs to the ligand-independent type I Hh signaling [17]. Abnormal Hh signaling and mutations in PTCH causing hyperactivation of Hh signaling have been observed in medulloblastoma [112,117,237]. In murine medulloblastoma cells, cyclopamine treatment suppressed proliferation in vitro and caused loss of neuronal stem-cell-like characteristics [238]. Also, itraconazole, a commonly used antifungal agent that works via inhibiting Hh signaling, was reported to inhibit the growth of medulloblastoma in a mouse allograft model, specifically through the inhibition of SMO [239]. Buonamici et al. reported that combination treatment of sonidegib with either PI3K inhibitor (NVP-BKM120) or the dual PI3K/mTOR inhibitor (NVP-BEZ235) significantly delayed medulloblastoma relapse [240]. Meanwhile, even though SMO inhibitor vismodegib (GDC-0449) was reported to have antitumor potency in early clinical studies of medulloblastoma, relapse appeared after treatment [154]. Indeed, the efficacy of vismodegib in combination with other drugs in treating medulloblastoma still remains unclear.

CD15 has been identified as a marker for medulloblastoma CSCs. Medulloblastoma with CD15^+^ cells expressed higher levels of Shh target genes, such as *GLI1* and *CYCLIN-D1*, than that with CD15^−^ cells [174,175]. The transcription factors GLI1/2 of the Hh pathway have been indicated to interact with stem-related factor NANOG. They bind to the specific cis-regulatory sequences of NANOG in both mouse and human stem cells [173]. When the Hh pathway was blocked by KAAD–cyclopamine, the expression of GLI1 and NANOG was reduced, and the self-renewal of medulloblastoma neurospheres was also significantly inhibited [173].

### 6.6. Leukemia

Hh pathway activity has a critical role in the maintenance of normal and neoplastic stem cells in the hematopoietic system and correlates with the chemotherapy-resistant phenotype of myeloid leukemia cells and disease recurrence [145,176,178]. However, Hh signaling has been reported to be less pronounced in chemosensitive cells [178]. In cancer with paracrine Hh signaling, such as pancreatic cancer, tumor cells secrete Hh ligands which then are received by remote cells in the stroma [198]. However, in leukemia, B-cell lymphomas, and multiple myelomas, the reverse paracrine signaling is reported. In this model, the Hh ligands are produced by the bone marrow stromal cells and are then received by the tumor cells [241]. These ligands support the proliferation and growth of surrounding cancerous tissue through upregulating the anti-apoptotic gene *Bcl2* [241].

In a chronic myelogenous leukemia (CML) study, Chen et al. reported that a loss of Smoothened (SMO) damaged the renewal of hematopoietic stem cells and impaired induction of CML by the BCR-ABL1 oncoprotein [145]. Moreover, Sonic hedgehog (SHH), smoothened (SMO), and GLI1 are significantly increased in CML patients compared with in normal people, suggesting that activation of the Hh pathway is a critical inducer of CML progression [242]. Hh signaling was also found to be implicated in multidrug resistance in CML. Treatment of cyclopamine along with nilotinib, a tyrosine kinase inhibitor (TKI), resulted in a re-sensitization to chemotherapy and inhibited CML progression [178]. Similar to solid cancer, leukemia cells also contain a subpopulation of leukemia-initiating cells or leukemic stem cells (LSCs) that have been reported to have the capability to self-renew and drive hematopoietic malignancies [145,243]. Hh signaling was activated through upregulation of SMO in LSCs and pharmacological inhibition of SMO reduced LSCs in vivo and delayed relapse after the end of treatment [176]. Indeed, loss of SMO causes depletion of CML stem cells while the activation of SMO supplements CML stem cell number and accelerates the disease without an impairment of normal hematopoiesis [145,176]. Additionally, GLI1, a downstream target of Hh signaling, was significantly upregulated in CD34^+^ chronic phase (CP) CML cells compared to in normal CD34^+^ hematopoietic cells [145]. Tyrosine kinase inhibitors (TKIs) such as imatinib, nilotinib, and dasatinib have been shown to have a dramatic effect on the eradication of chronic-phase (CP) CML. However, quiescent CML stem cells were not effectively targeted by TKIs. Drugs targeting both bulk progenitor and stem cell populations have been investigated for several decades. Irvine et al. showed that a combination of Hh inhibitor sonidegib (LDE225) and nilotinib succeeded in eradicating both CML LSCs and progenitor cells [177]. Cyclopamine and nilotinib, as well as an SMO antagonist (PF-04449913) and dasatinib, also showed effective elimination of CML LSCs [176,179]. These findings show that the combination of Hh inhibitors and TKIs reduces LSC self-renewal and clinically improves survival [177].

In acute myeloid leukemia (AML), activation of Hh signaling has been shown in several human AML cells, especially primary CD34^+^ leukemic cells [181]. Not only expression levels but also epigenetic modifications of Hh signaling components have been observed to correlate with disease status in AML patients [244]. A high level of GLI1 expression resulted in poor prognosis for AML patients [245]. Moreover, Hh pathway activation increased both the survival and drug resistance capability of CD34^+^ leukemia cells. Inhibition of Hh signaling with Smoothened antagonist cyclopamine, endogenous Hh inhibitor Hh-interacting protein, or anti-Hh neutralizing antibody induced apoptosis of CD34^+^ leukemic cells, although these CD34^+^ cell lines have a resistance to cytarabine (Ara-C) [181]. Furthermore, the combination of cyclopamine and Ara-C significantly reduced the drug resistance of CD34^+^ leukemic cells. Ara-C treatment in combination with cyclopamine significantly enhanced the sensitivity to Ara-C, which resulted in the complete abrogation of bulk leukemia cells [181]. PF-04449913 treatment modulated the cell cycle regulation and self-renewal signaling, which in turn reduced the leukemia initiation potential of AML cells and enhanced sensitivity to chemotherapy, overcoming the resistance in the bone marrow microenvironment [182]. Recently, Kakiuchi et al. reported that a novel SMO inhibitor, PF-913, impaired the self-renewal signature and cell cycle regulation of leukemic-stem-cell-like properties from AML patients, suggesting that PF-913 induced cell cycling in quiescent leukemia stem cells [180].

In lymphoblastic leukemia, Hh pathway components are upregulated in precursor B-ALL, and SMO inhibitor has been reported to reduce self-renewal of ALL cells in vitro and in vivo [173,184]. Moreover, mutations of the Hh pathway have also been described in T-ALL [246,247,248,249]. PTCH1 mutations were the most common Hh signaling found in T-ALL patients [246]. A study using the zebrafish model indicated that PTCH1 mutations contributing to chemotherapy resistance in human T-ALL were driving mutations responsible for oncogenic transformation in T-ALL malignancy [246]. Treatment of SANT-1 or KAAD cyclopamine was reported to induce the cell death of T-ALL cells as well as to re-sensitize the cells to chemotherapy [183]. Treatment of cyclopamine or IPI-926 was shown to inhibit the expansion of B-ALL LSCs by reducing the capability of self-renewal [173,184]. In chronic lymphocytic leukemia (CLL), the expression of Hh signaling molecules, such as GLI1, GLI2, BCL2, and SUFU, was significantly upregulated and correlated with B-CLL progression [241]. Furthermore, cyclopamine treatment inhibited bone marrow stromal cells, which resulted in a lower level of survival of B-CLL cells, indicating an important role of Hh signaling in the survival of B-CLL cells [241].

## 7. Conclusions

The Hh pathway is evolutionarily conserved and essential for the regulation of normal development and differentiation in invertebrates and vertebrates. However, aberrant Hh pathway activation by either mutation or ligand overexpression is closely correlated with many types of human cancers. Thus, targeting the Hh pathway may provide therapeutic options for the treatment of numerous types of human cancers. Effective inhibition of the Hh signaling pathway can be obtained through several approaches, such as using anti-Hh antibodies to inhibit receptor activation or small-molecule antagonists that inhibit downstream Hh signaling, such as SMO and GLI1. Synthetic Hh antagonists are now available, and some are undergoing clinical evaluation (Table 3). In this review, we have discussed the importance of Hh signaling in several types of cancers, such as basal cell carcinoma (BCC), leukemia, and medulloblastoma, and in colon, breast, and pancreatic cancer. The contributions of Hh signaling to cancer progression are summarized in Figure 2. Several agents targeting the Hh signaling pathway to eradicate CSCs in the same cancer have been proposed to have anticancer activity either as stand-alone drugs or in combination with standard chemotherapy drugs, and some of them have entered clinical trials as novel treatment strategies to regulate CSC replication, survival, and differentiation. However, not many of them have been successful in the total eradication of CSCs. Moreover, the toxicity associated with these drugs is a major impediment to their applicability. Therefore, considering the substantial contributions of Hh signaling to tumorigenesis and drug resistance, understanding the fundamental Hh pathway regulation in a precise manner will help in the development of new therapeutics to tackle Hh-pathway-activated cancers and achieve better therapeutic outcomes for cancer therapy.

## Figures and Tables

**Figure 1 cells-07-00208-f001:**
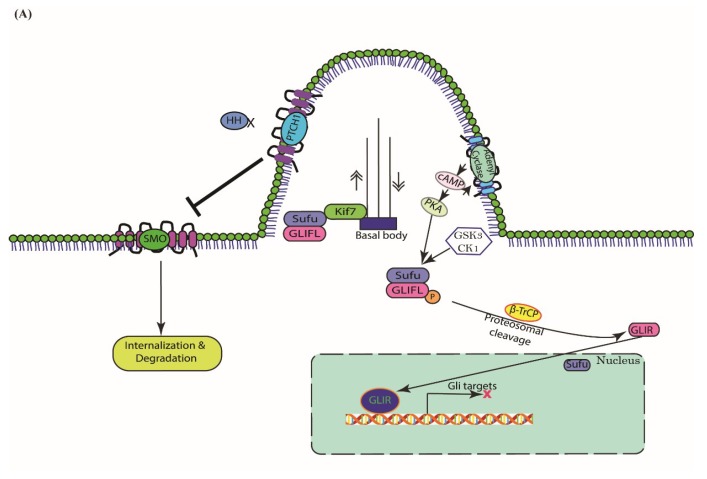
Schematic representation of the mammalian canonical Hh signaling pathway. (**A**) In the absence of Hh, PTCH1 localizes in the cilia and thus prevents the membrane localization and activation of SMO. SMO then internalizes in the membrane of intracellular endosomes and degrades. The full-length GLI (GLIFL) is modified by protein kinases protein kinase A (PKA), glycogen synthase kinase-3 (GSK3), casein kinase 1 (CK1), and the E3 ubiquitin ligase β-TrCP, which is then proteolytically cleaved into the transcriptional repressor form GLIR. The active form of GLI (GLIA) is suppressed by SUFU. After that, GLIR translocates to the nucleus and inhibits the expression of its target genes. (**B**) Binding of Hh to PTCH1 results in its internalization as well as the ciliary translocation and activation of SMO. GLIFL maintains its full length and bypasses the phosphorylation by PKA, GSK3, and CK1, which leads to the formation of activated GLI (GLIA). GLIA then translocates to the nucleus where it induces the expression of Hh target genes. Interestingly, Kif7 not only participates in the movement of GLIFL within the cilia but also inhibits Sufu which usually suppresses the formation of GLIA from GLIFL.

**Figure 2 cells-07-00208-f002:**
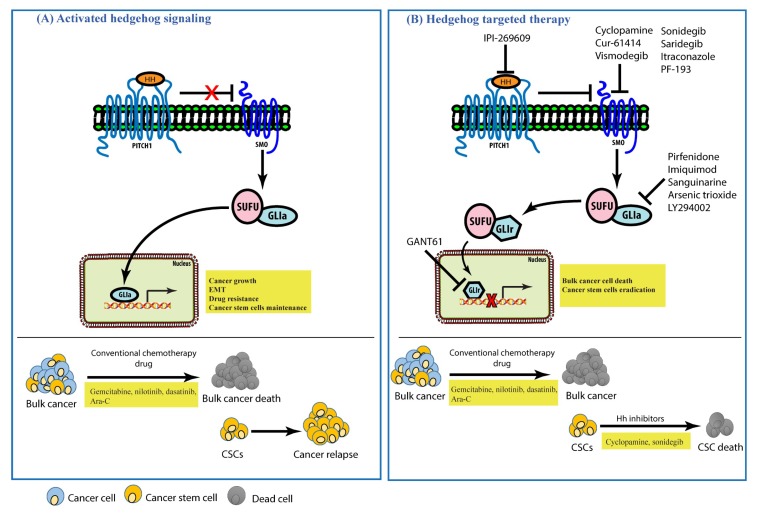
Schematic model of Hh activation and the therapeutic strategy for cancers. (**A**) In the presence of the Hh ligand, PTCH1 is internalized and SMO is activated, leading to the localization of GLI-1 in the nucleus. Gli-1 is activated and promotes the transcription of its target genes, which is important for the growth of cancer cells, promotion of epithelial–mesenchymal transition (EMT) mechanisms, drug resistance, and the maintenance of CSC population (upper). Conventional chemotherapy eliminates bulk cancer cells. However, residing CSCs escape from the therapy and trigger cancer relapse (bottom). (**B**) Inhibition by either Hh ligand, SMO, or GLI1 inhibitors led to the inactivation of Hh signaling. Gli-1 is suppressed by SUFU and fails to localize in the nucleus and activate Hh target genes (upper). Inhibition of Hh signaling together with conventional chemotherapeutics results in the death of cancer bulk cells and eradication of the cancer stem cell population which has escaped from conventional chemotherapeutics (bottom).

**Table 1 cells-07-00208-t001:** Hedgehog signaling activation in cancers.

Type	Characteristic	Cancer Type
**Type I**	Ligand independent	Basal cell carcinoma Medulloblastoma (MB) Pediatric brain tumor & Rhabdomyosarcoma
**Type II**	Ligand-dependent autocrine/juxtacrine signaling	Colorectal Prostate Liver Breast Ovarian Brain Melanoma
**Type III**	Ligand-dependent paracrine signaling	Pancreatic Leukemia (reverse paracrine)

**Table 2 cells-07-00208-t002:** Hedgehog signaling inhibition in cancer stem cells (CSCs).

Tumor Type	CSC Marker	Hh Inhibitors	Combination Therapy	References
Basal cell carcinoma (BCC)	CD200^+^CD45^−^	Smo antagonist	CD200^+^ neutralizing antibody	[164]
Colon cancer	CD133^+^	PTCH1 inhibitor (RU-SKI 43), Cyclopamine		[97,128]
Breast cancer	Lin^−^CD44^+^CD24^−^	Cyclopamine		[144,165]
Pancreatic cancer	CD44^+^CD24^+^ESA^+^	Sulforaphane, Baicalein, Sangunarine, GANT61	Gemtacibine and cyclopamine, cyclopamine derivates (CyT) +2gy irradiation	[166,167,168,169,170,171,172]
Medulloblastoma	CD15^+^, Sox2^+^	KAAD-cyclopamine		[173,174,175]
Chronic myelogenous leukemia (CML)	Lin^−^Sca1^+^cKit^+^CD34^+^	Sonidegib, cyclopamine, PF-04449913	Sonidegib and nilotinib, cyclopamine and nilotinib, PF-04449913 and dasatinib	[176,177,178,179]
Acute myelogenous leukemia (AML)		PF-04449913, cyclopamine, PF-913	Cyclopamine and Ara-C	[180,181,182]
Acute lymphoblastic leukemia (ALL)		Cyclopamine, IPI-926, KAAD-cyclopamine, SANT1		[183,184,185]

**Table 3 cells-07-00208-t003:** Hedgehog signaling inhibitors in clinical trials. Data from www.clinicaltrials.gov.

Compound	Target	Conditions	Phase	NCT Number	Combination Drug	Locations	Status
GDC-0449 (Vismodegib/Erivedge)	SMO	Basal cell carcinoma	Early Phase 1	NCT01631331	-	US	Completed
			Phase 1	NCT02639117	-	US	Completed
			Phase 2	NCT01543581	-	-	Completed
				NCT00833417	-	US	Completed
				NCT01201915	-	US	Completed
				NCT03035188	-	Germany	Recruiting
				NCT02667574	-	France	Active
				NCT01815840	-	US	Completed
				NCT01367665	-	Australia, Argentina	Completed
				NCT01898598	-	US	Terminated
				NCT02067104	-	US	Active
				NCT01700049	-	US	Active
				NCT01835626	-	US	Recruiting
				NCT01556009	-	US	Completed
				NCT02956889	-	Italy	Recruiting
				NCT00959647	FOLFOX, FOLFIRI, Bevacizumab	US	Completed
			Phase 4	NCT02436408	-	US	Recruiting
			Phase 1, 2	NCT02690948	Pembrolizumab	US	Active
			-	NCT01160250	-	US	Approved
				NCT02371967	-	Sweden	Active
				NCT02781389	-	Germany	Active
				NCT02674009	-	Germany	Active
				NCT02438644	-	-	Completed
		Colon cancer	Phase 2	NCT00636610	Bevacizumab, Modified FOLFOX, FOLFIRI	-	Completed
				NCT00959647	FOLFOX, FOLFIRI, bevacizumab	US	Completed
		Breast cancer	Phase 2	NCT02694224	Paclitaxel, Epirubicin, Cyclophosphamide	Spain	Recruiting
		Pancreatic cancer	Early Phase 1	NCT01713218	gemcitabine	Belgium	Unknown
			Phase 1	NCT00878163	erlotinib hydrochloride, gemcitabine	US	Active
				NCT01537107	sirolimus	US	Suspended
			Phase 2	NCT01088815	Gemcitabine, nab-Paclitaxel	US	Unknown
				NCT01195415	Gemcitabine Hydrochloride	US	Completed
			Phase 1, 2	NCT01064622	gemcitabine hydrochloride	US	Completed
		Medullo blastoma	Phase 1	NCT00822458	-	US	Completed
			Phase 2	NCT00939484	-	US	Completed
				NCT01239316	-	US	Completed
			Phase 1, 2	NCT01601184	Temozolomide	France	Recruiting
		Leukemia	Phase 2	NCT01880437	cytarabine	US	Terminated
				NCT01944943	-	France	Terminated
				NCT02073838	Ribavirin, Decitabine	US	Completed
LDE225 (Erismodegib/Sonidegib/Odomzo)	SMO	Basal cell carcinoma	Phase 1	NCT01208831	-	Hong Kong, Japan, Taiwan	Completed
				NCT00880308	-	US, Spain	Completed
			Phase 2	NCT01327053	-	US	Active
				NCT02303041	Buparlisib	US	Completed
				NCT01033019	-	Austria, Australia	Terminated
				NCT03534947	Imiquimod	Australia	Not yet
				NCT00961896	-	Austria, Switzerland	Completed
				NCT01350115	-	Austria, Belgium, Canada	Completed
			Phase 2, 3	NCT03070691	-	Belgium, Canada	Withdrawn
			-	NCT01529450	-	US	Completed
		Breast cancer	Phase 1	NCT02027376	Docetaxel	Spain	Unknown
			Phase 2	NCT01757327	-	-	Withdrawn
		Pancreatic cancer	Early Phase 1	NCT01694589	-	US	Withdrawn
			Phase 1	NCT01487785	gemcitabine	US	Completed
				NCT01485744	Fluorouracil, Leucovorin, Oxaliplatin, Irinotecan	US	Active
			Phase 1, 2	NCT02358161	nab paclitaxel	Netherland	Unknown
			-	NCT01911416	-	US	Withdrawn
		Medullo blastoma	Phase 1	NCT01208831	-	Hong Kong, Japan, Taiwan	Completed
				NCT00880308	-	Spain, US	Completed
			Phase 2	NCT01708174	-	US	Completed
			Phase 1, 2	NCT01125800	-	US	Completed
		Leukemia	Phase 1	NCT01456676	nilotinib	Canada, France, Germany	Completed
				NCT02129101	Azacitidine, Decitabine	US	Active
			Phase 2	NCT01826214	-	Australia, US	Completed
IPI-926 (Saridegib)	SMO	Basal cell carcinoma	-	NCT01609179	-	US	Completed
		Pancreatic cancer	Phase 1	NCT01383538	5-fluorouracil, Leucovorin, Irinotecan, Oxaliplatin	US	Completed
			Phase 1, 2	NCT01130142	gemcitabine	US	Completed
PF-04449913 (Glasdegib)	SMO	Leukemia	Phase 1	NCT01546038	ARA-C, Decitabine, Daunorubicin, Cytarabine	US	Active
				NCT02038777	ARA-C, Daunorubicin, Cytarabine, Azacitidine	Japan	Recruiting
				NCT02367456	Azacitidine	US	Recruiting
				NCT00953758	-	US	Completed
			Phase 2	NCT01841333	-	US	Recruiting
				NCT01842646	-	US	Active
				NCT03390296	Avelumab, azacitidine, utomilumab	US	Recruiting
			Phase 3	NCT03416179	-	US	Recruiting
BMS-833923	SMO	Leukemia	Phase 2	NCT01357655	Dasatinib	US	Terminated
			Phase 1, 2	NCT01218477	Dasatinib	US	Completed

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
