# Peer review of "Hedgehog Signaling in Cancer: A Prospective Therapeutic Target for Eradicating Cancer Stem Cells"

_cells, 2018, doi:10.3390/cells7110208_

Round 1
Reviewer 1 Report
Specific comments to the authors
The authors Ita Novita Sari et al. of the submitted review with the title “Hedgehog signaling in cancer: A prospective therapeutic target for eradicating cancer stem cells” present an intensive and detailed overview of known findings of Hedgehog (Hh) signaling pathway in several types of human cancer in relationship to cancer stem cells and inhibition.
Overall, the manuscript gives interesting aspects of Hh signaling and their inhibitors in human disesases and cancer, whereby some aspects should be clarified by the authors before accepting the manuscript for publication as mentioned below.
# As mentioned in the title “…target for eradicating cancer stem cells” the authors should summarize the known findings in a separate table for improving the reading and understanding of the therapeutic potency of Hh signaling pathway and cancer stem cells in the treatment of human diseases and cancers.
# As the Hh signaling pathway is essentially involved in epithelial-mesenchymal-transition which is again linked to cancer stem cells, the author should discuss this mechanistic link for therapeutic issues (see e.g. Mol Cell Biochem. 2014 Nov;396(1-2):257-68 and Oncol Lett. 2013 Jan;5(1):127-134. ).
# Furthermore, the problem of cancer cells surviving after chemotherapeutic treatment should be discussed by the authors (see Int J Oncol. 2012 Dec;41(6):2093-102.).
# Finally, the authors should try to clarify how parallel, sequential or iterative inhibition of Hh and cancer stem cells could improve the treatment of human diseases and cancers in future and which way should be chosen to find the best combinatory treatment scheme (perhaps in a new figure).
Author Response
Overall, the manuscript gives interesting aspects of Hh signaling and their inhibitors in human disesases and cancer, whereby some aspects should be clarified by the authors before accepting the manuscript for publication as mentioned below.
: We appreciate the comments
# As mentioned in the title “…target for eradicating cancer stem cells” the authors should summarize the known findings in a separate table for improving the reading and understanding of the therapeutic potency of Hh signaling pathway and cancer stem cells in the treatment of human diseases and cancers.
: In response to the comments, we generated 2 additional tables and put sentences in the manuscript as shown below (in blue color).
Page 11. CD133+ was identified as a marker of colon CSCs, and Hh-GLI1 signaling was also reported to play important roles in the colon CSCs survival and expansion [128]. Treatment of cyclopamine showed a reduction of the colon cancer progression and metastasis, and consistently inhibition of Hh signaling through GLI1-targeted shRNA displayed a dramatically lower colony forming ability of colon CSC [128].
Page 12. In breast cancer, Hh signaling was dependent on the autocrine manner of the Hh ligands of the tumor cells (type II Hh signaling).
Page 13. In pancreatic cancer, CD44+CD24+ESA+ cells were identified as a population of pancreatic CSC [213].
Page 14. Inhibition of Hh-GLI1 signaling by GANT-61 (GLI1 antagonist) showed an inhibition of pancreatic CSCs growth in both in vitro and in vivo model [219].
Page 14. Gemcitabine-resistant pancreatic cancer cells displayed a high expression of Hh signaling components as well as CSC markers [222]. Treatment of the cyclopamine into these resistant cells reverted the acquired chemoresistance and decreased the expression of CSC markers [222, 223]. In addition, treatment of cyclopamine derivate (CyT) along with the focal irradiation showed an additive effect on tumor sphere formation in vitro as well as inhibition of the lymph node metastasis in in vivo model [224]. Taken together, targeting Hh signaling in pancreatic CSCs showed a better outcome in the ablation of pancreatic cancer cells.
Page 15. Hh signaling was also found to be implicated in multidrug resistance in CML. Treatment of cyclopamine along with the nilotinib, a tyrosine kinase inhibitor (TKI) resulted in a re-sensitization to the chemotherapy and inhibited CML progression [233]
Page 15. Cyclopamine and nilotinib, as well as SMO antagonist (PF-04449913) and dasatinib, also showed the effective elimination of CML LSCs [232, 238].
Page 16. Treatment of SANT-1 or KAAD cyclopamine was reported to induce the cell death of T-ALL cells as well as re-sensitize the cells to the chemotherapy [249]. Treatment of cyclopamine or IPI-926 was shown to inhibit the expansion of B-ALL LSCs by reducing the capability of self-renewal. [231, 244].
Page 29. Table 1. Hedgehog signaling activation in cancers comment 1
Type | Characteristic | Cancer type |
Type I |
Ligand-independent | Basal cell carcinoma Medulloblastoma (MB) Pediatric brain tumor & Rhabdomyosarcoma |
Type II |
Ligand-dependent autocrine/ juxtacrine signaling | Colorectal Prostate Liver Breast Ovarian Brain Melanoma |
Type III | Ligand-dependent paracrine signaling | Pancreatic Leukemia (reverse paracrine) |
Table 2. Hedgehog signaling inhibition in the cancer stem cells (CSCs) comment 1
Tumor type | CSC Marker | Hh inhibitors | Combination Therapy | Reference |
Basal cell carcinoma (BCC) | CD200+, CD45- | Smo antagonist | CD200+ neutralizing antibody | [179] |
Colon cancer | CD133+ | PTCH1 inhibitor (RU-SKI 43), Cyclopamine | [97, 128] | |
Breast cancer | Lin-CD44+CD24- | Cyclopamine | [144, 208] | |
Pancreatic cancer | CD44+CD24+ESA+ | Sulforaphane, Baicalein, Sangunarine, GANT61
| Gemtacibine and cyclopamine, cyclopamine derivates (CyT) +2gy irradiation
| [217, 219-224] |
Medulloblastoma | CD15+, Sox2+ | KAAD-cyclopamine | [229-231] | |
Leukemia CML |
Lin-Sca1+cKit+CD34+ | Sonidegib, cyclopamine, PF-04449913 | Sonidegib and nilotinib, cyclopamine and nilotinib, PF-04449913 and dasatinib | [232, 233, 237, 238] |
AML | PF-04449913, cyclopamine, PF-913 | Cyclopamine and Ara-C, | [239, 242, 243] | |
ALL | Cyclopamine, IPI-926, KAAD-cyclopamine, SANT1 | [244, 249, 250] |
# As the Hh signaling pathway is essentially involved in epithelial-mesenchymal-transition which is again linked to cancer stem cells, the author should discuss this mechanistic link for therapeutic issues (see e.g. Mol Cell Biochem. 2014 Nov;396(1-2):257-68 and Oncol Lett. 2013 Jan;5(1):127-134. ).
: We added new sentences as shown below.
Page 9. Moreover, accumulating evidence suggests that hedgehog signaling plays a role in the properties of CSCs in several cancers. For instance, the expression of hedgehog signaling components such as PTCH1, GLI1, and GLI2 is up-regulated in normal human mammary stem/progenitor cells of mammosphere and the Hh signaling pathway is induced in human breast CSCs [144]. Also, Hh pathway is critical for the maintenance and self-renewal of normal and neoplastic stem cells of the hematopoietic system, including CSCs in myeloid leukemia [145] as well as in solid cancers such as melanoma and pancreatic cancer [146, 147]. Some cancer cells surviving after chemotherapeutic treatment such as gemcitabine harbor stem cell-like properties and display characteristics associated with epithelial-mesenchymal transition (EMT) [148]. Surprisingly, Hh signaling also controls the expression of ABC transporter proteins such as multi-drug resistance protein-1, leading to the chemoresistance of CSCs [149]. Besides, as CSCs are known to undergo EMT, Hh pathway inhibitors such as cyclopamine and Gant-61 [150] or panobinostat [151] can be used as a potential novel treatment strategy to suppress EMT process and distant metastasis in cancers
# Furthermore, the problem of cancer cells surviving after chemotherapeutic treatment should be discussed by the authors (see Int J Oncol. 2012 Dec;41(6):2093-102.).
: We discussed the point as shown below.
Page 9. Moreover, accumulating evidence suggests that hedgehog signaling plays a role in the properties of CSCs in several cancers. For instance, the expression of hedgehog signaling components such as PTCH1, GLI1, and GLI2 is up-regulated in normal human mammary stem/progenitor cells of mammosphere and the Hh signaling pathway is induced in human breast CSCs [144]. Also, Hh pathway is critical for the maintenance and self-renewal of normal and neoplastic stem cells of the hematopoietic system, including CSCs in myeloid leukemia [145] as well as in solid cancers such as melanoma and pancreatic cancer [146, 147]. Some cancer cells surviving after chemotherapeutic treatment such as gemcitabine harbor stem cell-like properties and display characteristics associated with epithelial-mesenchymal transition (EMT) [148]. Surprisingly, Hh signaling also controls the expression of ABC transporter proteins such as multi-drug resistance protein-1, leading to the chemoresistance of CSCs [149]. Besides, as CSCs are known to undergo EMT, Hh pathway inhibitors such as cyclopamine and Gant-61 [150] or panobinostat [151] can be used as a potential novel treatment strategy to suppress EMT process and distant metastasis in cancers
# Finally, the authors should try to clarify how parallel, sequential or iterative inhibition of Hh and cancer stem cells could improve the treatment of human diseases and cancers in future and which way should be chosen to find the best combinatory treatment scheme (perhaps in a new figure).
: We edited figure 2 to clarify the point.

Reviewer 2 Report
- The manuscript is well written and prepared and a pleasure to read, I have some points and suggestions though:
- The manuscript would benefit from some more figures, there is a nice figure 1 and 2, but I feel that of some of the concepts presented are quite complicated for the uninitiated reader and would benefit from further illustration.
- Focussing on the role of Hedgehog in cancer ignores that stem cell renewal is pivotal as an antagonist of ageing and recovery following chemo/radiotherapy. This the other side of the coin so to speak and I think it would be good if the authors mention explore this a little better in the manuscript or refer to a review on this subject.
- Hh proteins (line 47 and below). Expression data now focusses on embryogenesis only. There a plethora of data, however, that in normal physiology Hh remains expressed during life (some of this data is being referred to but the concept is not well-developed in the text) and I think this point is relevant in the context of this manuscript.
- Non-canonical Hedgehog signaling appears quite important for the effects of Hedgehog described in this manuscript (see also ref 150), but is not covered too well in comparison to the canonical signaling. This section should better describe effects on arachidonic metabolism (leukotrienes etc) and chemotaxis and possibly also other kinase effects.
- Different types of Hedgehog dependent cancer deserve an introduction as to where this classification derives from (and possibly an illustration).
- The dependency of some cancers on hedgehog ligands, even if local hedgehog production is absent suggests that long-range Hedgehog signaling is important in this respect. No too much is known in this respect, but the authors should discuss potential (bloodborn?) sources of this longrange hedgehog signaling (surprising in view of the hydrophobic nature of the ligand)
- A potential fourth type of effect of Hedgehog in cancer is angiogenesis, what are the thoughts of the authors in this respect?
- Smo inhibitor therapy of cancer may suffer from the development of resistance. Some words/thoughts on this issue, or at least a reference to a review might benefit this manuscript
Author Response
- The manuscript is well written and prepared and a pleasure to read, I have some points and suggestions though:
: We appreciate the comment.
- The manuscript would benefit from some more figures, there is a nice figure 1 and 2, but I feel that of some of the concepts presented are quite complicated for the uninitiated reader and would benefit from further illustration.
: We edited figure 2 to clarify this point.
- Focussing on the role of Hedgehog in cancer ignores that stem cell renewal is pivotal as an antagonist of ageing and recovery following chemo/radiotherapy. This the other side of the coin so to speak and I think it would be good if the authors mention explore this a little better in the manuscript or refer to a review on this subject.
: We discussed this part as shown below in blue color.
Page 9. Moreover, accumulating evidence suggests that hedgehog signaling plays a role in the properties of CSCs in several cancers. For instance, the expression of hedgehog signaling components such as PTCH1, GLI1, and GLI2 is up-regulated in normal human mammary stem/progenitor cells of mammosphere and the Hh signaling pathway is induced in human breast CSCs [144]. Also, Hh pathway is critical for the maintenance and self-renewal of normal and neoplastic stem cells of the hematopoietic system, including CSCs in myeloid leukemia [145] as well as in solid cancers such as melanoma and pancreatic cancer [146, 147]. Some cancer cells surviving after chemotherapeutic treatment such as gemcitabine harbor stem cell-like properties and display characteristics associated with epithelial-mesenchymal transition (EMT) [148]. Surprisingly, Hh signaling also controls the expression of ABC transporter proteins such as multi-drug resistance protein-1, leading to the chemoresistance of CSCs [149]. Besides, as CSCs are known to undergo EMT, Hh pathway inhibitors such as cyclopamine and Gant-61 [150] or panobinostat [151] can be used as a potential novel treatment strategy to suppress EMT process and distant metastasis in cancers
- Hh proteins (line 47 and below). Expression data now focusses on embryogenesis only. There a plethora of data, however, that in normal physiology Hh remains expressed during life (some of this data is being referred to but the concept is not well-developed in the text) and I think this point is relevant in the context of this manuscript.
: We added more information as suggested.
Page 3. Hh proteins are one of the secreted signaling protein families that can be diffused and establish gradients in target tissues after secretion, and thus are responsible for cell interactions during embryonic development [26]. Hh proteins can act as either morphogens or mitogens which control developmental processes during multiple developmental stages in many different tissues, in a dose-dependent manner [26]. Hh signaling plays an important role in the postnatal bone homeostasis [27, 28] as well as regulating the development and function of theca cells [29] and Leydig cells [30], indicating that Hh ligands not only participate in the embryonic patterning but are also involved in the maintenance of tissue homeostasis during lifetime.
- Non-canonical Hedgehog signaling appears quite important for the effects of Hedgehog described in this manuscript (see also ref 150), but is not covered too well in comparison to the canonical signaling. This section should better describe effects on arachidonic metabolism (leukotrienes etc) and chemotaxis and possibly also other kinase effects.
: We put more information as shown below in blue color.
Page 6-7. However, emerging evidence over the past few years has indicated that Hh proteins can also signal through GLI-independent mechanisms, the so-called “non-canonical” Hh signaling [5]. In this pathway, all the responses to Hh ligands are not through transcriptional changes mediated by the transcription factors of the Gli family [96]. Interestingly, Hedgehog signaling in CSCs is non-canonical PTCH1-dependent, which is a positive regulator of WNT signaling whereas canonical SMO-dependent Hedgehog signaling is involved in the downregulation of WNT signaling in normal and differentiated cancer cells through nuclear localization of GLI1 [97] (comment 8). Non-canonical Hh signaling has been classified into two types: 1) SMO-independent and exclusively through PTCH, and 2) SMO-downstream, mediated by activation of small GTPases, but GLI transcription factors-independent. SMO-independent GLI activation pathway has additionally been shown to be noncanonical Hh signaling [98, 99].
The SMO-independent noncanonical signaling pathway, which operates via PTCH and three Hh proteins, SHH, IHH and DHH, can inhibit the activation of caspase-3 and promote the cell survival through inactivating the PTCH1 pro-apoptotic activity in an SMO-independent manner [100]. Surprisingly, it is indicated that the autocrine Shh-Ptch-Smo signaling is not required in pancreatic ductal cells for the progression of pancreatic ductal adenocarcinoma (PDAC), as the deletion of SMO in the pancreas does not affect the multistage development of PDAC tumors [98]; instead, GLI1 is regulated by TGF-β and KRAS, and Gli-NF-κB oncogenic activation is required for the Kras-dependent transformation of cultured PDAC cancer cells [101]. Besides, Smo- and Gli-independent Hedgehog also participate in the regulation of cell cycle through Cyclin B1 and G-protein receptor kinase-2 (GRK2) [102] (comment 8).
In the noncanonical signaling pathway, which works through functions or activities of SMO, it has been indicated that via the rapid stimulation of Rac1 and RhoA small GTPases by SMO, heterotrimeric Gi Proteins are required for SHH-induced fibroblast migration in human endothelial cells [103]. Additionally, cholangiocarcinoma cells exhibit a GLI- and cilia-independent non-canonical Hh signaling pathway, which contributes to the mammalian tumor cell engraftment and chemotaxis, including cytoskeleton remodeling and cell migration [104]. Moreover, arachidonic acid (ARA) metabolism through lipoxygenase pathway plays an important role in Gli-independent Hh signaling in fibroblast migration [105] and neuronal development; being particular in the synthesis of leukotrienes [106]. In addition, noncanonical Hh signaling pathway is also involved in controlling the axon guidance in a Smo-dependent manner by inducing phosphorylation and activation of Src family kinases (SFKs) to alter axon trajectories [107] (comment 8).
- Different types of Hedgehog dependent cancer deserve an introduction as to where this classification derives from (and possibly an illustration).
We generated a new table and put some sentences.
Page 9. It is believed that the upregulation of Hh signaling is the ace and driver of all BCC malignancies. Hh signaling in the BCC belongs to the ligand-independent type I Hh signaling [65].
Page 11. The Hedgehog (Hh) signaling pathway is essential for cell growth as well as the development of gastrointestinal tracts [181]. Accumulating evidence suggested that Hh signaling plays important roles in the development of colon cancer [182, 183]. Hh signaling in the CRC belongs to the ligand-dependent type II Hh signaling [128].
Page 29. Table 1. Hedgehog signaling activation in cancers
Type | Characteristic | Cancer type |
Type I |
Ligand-independent | Basal cell carcinoma Medulloblastoma (MB) Pediatric brain tumor & Rhabdomyosarcoma |
Type II |
Ligand-dependent autocrine/ juxtacrine signaling | Colorectal Prostate Liver Breast Ovarian Brain Melanoma |
Type III | Ligand-dependent paracrine signaling | Pancreatic Leukemia (reverse paracrine) |
- The dependency of some cancers on hedgehog ligands, even if local hedgehog production is absent suggests that long-range Hedgehog signaling is important in this respect. No too much is known in this respect, but the authors should discuss potential (bloodborn?) sources of this longrange hedgehog signaling (surprising in view of the hydrophobic nature of the ligand)
: We discussed this part as shown below.
Page 3. Hh proteins undergo maturation by processing post-translationally and multiple covalent modifications before the active ligand is released and activates the Hh pathway [28]. N-terminal signal sequence is removed after being translated and the Hh protein is then autocatalytically cleaved internally [31]. During this process, the C-terminal domain acts as an intramolecular cholesterol transferase to promote the covalent attachment of a molecule of cholesterol to the N -terminal signaling domain [32] while a palmitic acid moiety is added to the N-terminal cysteine residue by the acyltransferase Skinny Hedgehog (Ski) to form a mature Hh protein dually modified by cholesteryl and palmitoyl adducts [33]. Surprisingly, this unusual dual lipid modification plays a vital role in how Hh ligands could move far away from the organizing centers where Hh is specifically expressed and acts as a morphogen [34, 35]. Therefore, this dual lipid modification could facilitate the control of the long-range activity of Hh ligands in cancers even where the local Hh proteins are limited [36]
- A potential fourth type of effect of Hedgehog in cancer is angiogenesis, what are the thoughts of the authors in this respect?
: We added more information as shown below.
Page 9. Interestingly, recent evidence indicated that Hh signaling plays an essential role in tumor angiogenesis, through VEGF-A induction in stromal perivascular cells [152], particularly in triple-negative breast cancer [153].
- Smo inhibitor therapy of cancer may suffer from the development of resistance. Some words/thoughts on this issue, or at least a reference to a review might benefit this manuscript.
We mentioned the resistance of inhibitors below.
Page 9. Thus, many inhibitors of Hh pathway have been developed and tried clinically with promising results (Table 2 and 3). However, emerging data indicate that cancer cells may also develop resistance to Hh inhibitors [154-159]. For example, in the medulloblastoma and BCC, treatment of vismodegib (GDC-0449) and sonidegib (LDE-225), two drugs that have been shown a significant result in the clinical trials, caused the acquired drug resistance in the residing cells of these cancers [154, 155]. Using genomic analysis, Sharpe et al. indicated that acquired drug resistant cells employed SMO-mutations in BCC, which further drove drug resistance to SMO inhibitors [155]. The SMO-D4738H mutation was found in 42.5% of BCC patients that acquired drug resistance [156]. This mutation blocked the binding of vismodegib and sonigedib to the mutant SMO protein [156]. While the new generation of SMO inhibitors is undergoing clinical trials, a recent study by Li et al. demonstrated that benzimidazole derivates named HH-1, HH-3, and HH-20 are potent novel SMO-inhibitors [160-163]. Moreover, HH-13 and HH-20 showed a prompt inhibition on acquired drug resistance SMO—D37738H cells as well [163].

Reviewer 3 Report
The review is well written and covers the role of Hedgehog signalling in a wide range of tumours, but omits to discuss the impact of Hedgehog signalling on T-cell immunity and its relevance to cancer.
Hedgehog pathway activation in T-cells has been shown to reduce T-cell activation (Rowbotham et al, Blood 2007; Furmanski et al J. Cell Science 2015), whereas inhibition of Hedgehog-mediated signalling has been shown to increase T-cell activation and Ifng expression (Rowbotham et al Cell Cycle 2008; Furmanski et al JI 2013), indicating that Hh-inhibitors could increase the anti-tumour immune response. Indeed, Hh-inhibitors have been shown to promote active immune responses in BCC (Otsuka et al, Clin Cancer Res 2015). The impact of Hh to dampen T-cell function should be discussed and these papers cited.
In the section,' Hh proteins' the review also omitted important functions of Dhh in relation of spleen and bone marrow haematpoiesis/erythropoiesis (Lau et al, Blood 2012), and the known regulatory functions of Shh (Shah et al, JI 2004; Solanki et al Development 2018) and Ihh (Outram et al Blood 2009) in T-cell development; and Shh in B-cell development (Solanki et al JEM 2017). This should be added to the section.
Author Response
The review is well written and covers the role of Hedgehog signalling in a wide range of tumours, but omits to discuss the impact of Hedgehog signalling on T-cell immunity and its relevance to cancer.
: We appreciate the comment.
Hedgehog pathway activation in T-cells has been shown to reduce T-cell activation (Rowbotham et al, Blood 2007; Furmanski et al J. Cell Science 2015), whereas inhibition of Hedgehog-mediated signalling has been shown to increase T-cell activation and Ifng expression (Rowbotham et al Cell Cycle 2008; Furmanski et al JI 2013), indicating that Hh-inhibitors could increase the anti-tumour immune response. Indeed, Hh-inhibitors have been shown to promote active immune responses in BCC (Otsuka et al, Clin Cancer Res 2015). The impact of Hh to dampen T-cell function should be discussed and these papers cited.
: In response to the comment, we put more information as shown below in blue color.
Page 10. Besides the function in the promotion of tumorigenesis, Hh signaling has also been reported to be involved in BCC tumor microenvironment, especially cancer immunity [165]. A study by Otsuka et al. showed that Hh signaling inhibition led to the enhanced adaptive immune responses in BCC [165]. Hh components were known to be highly expressed in the thymus and its upregulation led to the inhibition of T cell activation [166-168]. IL-4 upregulation in the tumor microenvironment was reported to enhance tumorigenesis as well as inhibit antitumor response [169, 170]. Furmanski et al. also showed that IL-4 is a transcriptional target of Hh signaling in T cells [171]. In addition, Hh signaling inhibition was also shown to increase interferon-gamma (IFNγ) expression [171, 172]. Taken together, it suggests that targeting Hh signaling in the BCC would increase the effectiveness of the cancer therapy by modulating the antitumor activity of the immune cells.
In the section,' Hh proteins' the review also omitted important functions of Dhh in relation of spleen and bone marrow haematpoiesis/erythropoiesis (Lau et al, Blood 2012), and the known regulatory functions of Shh (Shah et al, JI 2004; Solanki et al Development 2018) and Ihh (Outram et al Blood 2009) in T-cell development; and Shh in B-cell development (Solanki et al JEM 2017). This should be added to the section.
We added the information as suggested.
Page 4. Interestingly, SHH [54, 55] and IHH [56] were shown to be involved in the commitment and differentiation of the T cell lineage, as well as the proliferation and survival of developing T cells. It was also reported that SHH regulates the B-lineage commitment of hematopoietic progenitor cells and the development of B cells [57] (comment 14). Whereas SHH and IHH are closely related to each other, DHH is the closest homolog to D. melanogaster of all the discovered Hh ligands [5]. DHH expression is mainly restricted to gonads, such as Sertoli cells [58] and Leydig cells [30] in the testis and granulosa cells of growing follicles in the ovaries [29], where it plays an important role in the gametogenesis and steroidogenesis. Besides, DHH could negatively regulate erythrocyte differentiation at multiple stages in both the spleen and bone marrow [59] (comment 14).